# Plasma Proteome Signature to Predict the Outcome of Breast Cancer Patients Receiving Neoadjuvant Chemotherapy

**DOI:** 10.3390/cancers13246267

**Published:** 2021-12-14

**Authors:** Sungchan Gwark, Hee-Sung Ahn, Jeonghun Yeom, Jiyoung Yu, Yumi Oh, Jae Ho Jeong, Jin-Hee Ahn, Kyung Hae Jung, Sung-Bae Kim, Hee Jin Lee, Gyungyub Gong, Sae Byul Lee, Il Yong Chung, Hee Jeong Kim, Beom Seok Ko, Jong Won Lee, Byung Ho Son, Sei Hyun Ahn, Kyunggon Kim, Jisun Kim

**Affiliations:** 1Department of Surgery, Ewha Womans University Mokdong Hospital, Ewha Womans University College of Medicine, Seoul 07985, Korea; drgwark@ewha.ac.kr; 2Asan Institute for Life Sciences, Asan Medical Center, Seoul 05505, Korea; hsahn_2021@amc.seoul.kr (H.-S.A.); scarlet202@amc.seoul.kr (J.Y.); yum7102@amc.seoul.kr (Y.O.); 3Convergence Medicine Research Center, Asan Institute for Life Sciences, Asan Medical Center, Seoul 05505, Korea; nature8309@amc.seoul.kr; 4Department of Biomedical Sciences, University of Ulsan College of Medicine, Seoul 05505, Korea; 5Department of Oncology, Asan Medical Center, University of Ulsan College of Medicine, Seoul 05505, Korea; jaeho.jeong@amc.seoul.kr (J.H.J.); drjiny@amc.seoul.kr (J.-H.A.); khjung@amc.seoul.kr (K.H.J.); sbkim3@amc.seoul.kr (S.-B.K.); 6Department of Pathology, Asan Medical Center, University of Ulsan College of Medicine, Seoul 05505, Korea; backlila1@amc.seoul.kr (H.J.L.); gygong@amc.seoul.kr (G.G.); 7Department of Surgery, Asan Medical Center, University of Ulsan College of Medicine, Seoul 05505, Korea; newstar153@amc.seoul.kr (S.B.L.); doorkeeper1@amc.seoul.kr (I.Y.C.); heejeong_kim@amc.seoul.kr (H.J.K.); spdoctor@amc.seoul.kr (B.S.K.); leejw@amc.seoul.kr (J.W.L.); brdrson@amc.seoul.kr (B.H.S.); ahnsh@amc.seoul.kr (S.H.A.); 8Clinical Proteomics Core Laboratory, Convergence Medicine Research Center, Asan Medical Center, Seoul 05505, Korea; 9Bio-Medical Institute of Technology, Asan Medical Center, Seoul 05505, Korea

**Keywords:** liquid biopsy, breast cancer, neoadjuvant chemotherapy, proteome, LC-MS/MS

## Abstract

**Simple Summary:**

The prognostic impact of plasma protein biomarkers in breast cancer patients treated with neoadjuvant chemotherapy (NCT) was evaluated using a proteomics approach. Three biomarkers were identified among differentially expressed proteins. The plasma concentration of APOC3 was higher in the pathological complete response (pCR) group, whereas MBL2, ENG, and P4HB were upregulated in the non-pCR group. Univariate survival analysis was performed to identify protein biomarkers that could classify patients into low- and high-risk groups. The results showed that MBL2 and P4HB were statistically significantly associated with disease-free survival (log-rank test *p* < 0.05); P4HB was statistically significantly associated with overall survival (log-rank test *p* < 0.05), whereas MBL2 was statistically significantly associated with distant metastasis-free survival (log-rank test *p* < 0.05). The results demonstrated that protein markers from non-invasive liquid biopsy sampling correlate with pCR and survival in breast cancer patients receiving NCT. Further investigation of these protein markers may help clarify their role in predicting prognosis and thus their therapeutic potential for preventing metastasis.

**Abstract:**

The plasma proteome of 51 non-metastatic breast cancer patients receiving neoadjuvant chemotherapy (NCT) was prospectively analyzed by high-resolution mass spectrometry coupled with nano-flow liquid chromatography using blood drawn at the time of diagnosis. Plasma proteins were identified as potential biomarkers, and their correlation with clinicopathological variables and survival outcomes was analyzed. Of 51 patients, 20 (39.2%) were HR+/HER2-, five (9.8%) were HR+/HER2+, five (9.8%) were HER2+, and 21 (41.2%) were triple-negative subtype. During a median follow-up of 52.0 months, there were 15 relapses (29.4%) and eight deaths (15.7%). Four potential biomarkers were identified among differentially expressed proteins: APOC3 had higher plasma concentrations in the pathological complete response (pCR) group, whereas MBL2, ENG, and P4HB were higher in the non-pCR group. Proteins statistically significantly associated with survival and capable of differentiating low- and high-risk groups were MBL2 and P4HB for disease-free survival, P4HB for overall survival, and MBL2 for distant metastasis-free survival (DMFS). In the multivariate analysis, only MBL2 was a consistent risk factor for DMFS (HR: 9.65, 95% CI 2.10–44.31). The results demonstrate that the proteomes from non-invasive sampling correlate with pCR and survival in breast cancer patients receiving NCT. Further investigation may clarify the role of these proteins in predicting prognosis and thus their therapeutic potential for the prevention of recurrence.

## 1. Introduction

Neoadjuvant chemotherapy (NCT) provides several benefits for locally advanced breast cancer (BC) patients. Down-staging of tumors may increase the probability of breast conservation with better cosmesis [1,2,3]. Additionally, it allows in vivo monitoring of the response of tumors to therapy, which could be helpful for predicting pathological responses [4,5,6,7,8,9]. Tumors that respond well to a given therapy show better outcomes, and pathological complete response (pCR) is a surrogate factor for survival in the neoadjuvant setting [10,11,12]. However, the value of pCR for predicting prognosis in different subtypes of BC, especially in estrogen receptor (ER)-positive and HER2-negative tumors, is under debate. Because of substantial inconsistencies between clinical, radiological, and pathologic responses [13], extensive clinical/laboratory research has focused on achieving a more accurate prediction of the treatment response.

The Oncotype DX^®^ test provides genomic-based personalized information that enables the design of individualized treatments for patients with ER-positive primary BC [14]. The assay provides information on clinicopathological variables and has been validated for use in predicting the response to chemotherapy in ER-positive and HER2-negative BC. However, the assay requires a surgical specimen (formalin-fixed paraffin-embedded (FFPE) >10 × 10 µm) and two to three weeks of processing time, with costs higher than USD 4000. In addition, its indication is limited to the ER-positive and HER2-negative subtype. Plasma cancer biomarkers can provide useful information relevant to the diagnosis of several malignancies. In BC, several serum tumor biomarkers have been proposed, including the MUC-1 antigen (CA 15-3), the onco-fetal protein carcinoembryonic antigen (CEA), the oncoprotein HER-2/neu, and the cytokeratin tissue polypeptide specific antigen (TPS). However, because of their low sensitivity and specificity, the clinical utility of these serum markers is limited [15,16,17,18]. Compared with tissue biopsy, a quick and easy ‘liquid biopsy’ provides a less invasive and simpler method to assess tumor response through simple blood collection [19,20]. Circulating tumor cells (CTCs), which are detected by liquid biopsy, serve as a prognostic factor in metastatic BC, where a high CTC level is associated with poor prognosis [21,22]. However, analysis of CTCs in non-metastatic tumors has shown conflicting results [23,24,25,26,27], including in BC patients treated with NCT [28,29,30,31].

In translational research, proteomic-based BC research uses clinical samples such as plasma and tissues, to improve BC care through screening, early or companion diagnostics studies, predicting prognosis, subtyping, predicting metastasis and therapeutic responses, and discovering new drug targets [32,33,34,35,36,37]. Some protein signatures associated with the response to NCT were identified using FFPE tissues. PYCR1 and ALDH18A1 were identified by comparing the proteome of tissue before and after NCT with that of normal tissue [38]. The combination of four proteins, RAC2, RAB6A, BIEA, and IPYR, showed the best performance for predicting recurrence after NCT in triple-negative BC (TNBC) patients [39]. CD45 was identified as a predictor of pCR to neoadjuvant HER2-targeted therapy using the spatial proteomics approach [40]. The VEGF inhibition response predictor score, which is derived using nine-protein signatures, predicts the response to bevacizumab NCT in HER2-negative BC [41]. Analysis of blood samples identified differentially expressed proteins related to the pathological criteria for predicting patient responses to NCT [42,43,44,45,46], and blood exosomes [47] have been analyzed using proteomic approaches to predict the response to NCT.

Recent proteomic-centric multiomics studies analyzed BC clinical samples to classify BC into subtypes using protein and phosphorylated protein information, which led to the identification of novel subtypes such as basal-like and luminal B tumors by infiltration of immunological components [48]. Mass spectrometry-based proteomics that include analysis of post-translational modifications such as phosphorylation or acetylation, combined with next-generation DNA and RNA sequencing profiles, may provide a more comprehensive description of breast tumors [49,50,51]. Proteogenomic approaches highlight the potential of proteomics for clinical research in cancer through the identification of targetable signaling pathways and more precise curation of the biological signatures of tumor heterogeneity. Specifically, different genetic backgrounds may affect the inhibitory relationship between target kinases and tumor suppressors by post-translational modification.

Reliable biomarkers related to treatment response and survival in BC patients receiving NCT are currently lacking, and comorbidities are important for chemotherapy indication and regimen selection [52]. Proteome analysis may provide more reliable and direct information that can be used for monitoring the treatment response and for selecting specific treatment agents. Additionally, plasma proteome analysis can overcome the limitations of current biomarkers and tools by providing faster processing times (<2 days) and by using small samples (40 µL plasma). The aim of the present study was to identify potential biomarkers for predicting the response to therapy and for predicting recurrence by performing whole proteome analyses of the plasma of BC patients undergoing neoadjuvant systemic therapy.

## 2. Materials and Methods

### 2.1. Study Participants and Surveillance

Inclusion criteria for the baseline database of the prospective cohort were, (1) women aged >20 years, (2) pathologically diagnosed primary invasive breast cancer, (3) no distant metastasis at time of diagnosis (no de novo stage IV disease), (4) had undergone preoperative systemic treatment with curative intent, and (5) had agreed and signed consent. All patients had 4 mL blood drawn at time of diagnosis before chemotherapy. Among the 60 initial consecutively enrolled patients treated with NCT at Asan Medical Center in Seoul, Korea, between February 2014 and April 2017, 51 were eligible for final analysis with sufficient protein extracted for analysis. All patients’ survival outcomes were updated, including loco-regional recurrences, distant metastasis and death information. The study was approved by the Institutional Review Board (IRB) of Asan Medical Center (Seoul, Korea; IRB-e no. 2013-1048), and was performed in compliance with the REMARK criteria [53]. Written informed consent was obtained from all participants. All experiments were performed in accordance with the relevant guidelines and regulations.

All patients of the study received standard treatment, and regular surveillance was performed. The initial diagnostic and follow-up work-up included mammography, breast ultrasound imaging, magnetic resonance imaging, chest X-rays, blood sampling, and clinical examination. ER and progesterone receptor expression were evaluated based on the Allred score [54]. HER2 status was considered negative if the immunohistochemistry score was 1+, or if the score was 2+ and the result of fluorescence or silver in situ hybridization for HER2 amplification was negative [55]. Clinical and histopathological staging was based on the 7th edition of the Cancer Staging Manual of the American Joint Committee on Cancer [56]. The clinical treatment response was evaluated by both physical examination and imaging assessments at each treatment timeline (baseline, after the first treatment, and after completing the course of NCT). Tumor response was assessed by the Response Evaluation Criteria In Solid Tumors (RECIST 1.1) [57,58].

### 2.2. Sample Preparation

For the proteomic analyses, 51 clinical plasma samples were prepared for LC-MS analysis. Plasma samples were loaded onto a MARS14 column (100 × 4.6 mm; Agilent Technology, Palo Alto, CA, USA) on a Shimadzu binary HPLC system (20A Prominence; Shimadzu, Tokyo, Japan) in order to deplete 14 highly abundant proteins; the unbound fraction was lyophilized with a cold trap (CentriVap Cold Traps; Labconco, Kansas City, MO, USA). Dried samples were resuspended in 400 μL of 5% SDS in 50 mM TEAB (pH 7.55), and dithiothreitol was added to a final concentration of 20 mM for 10 min at 95 °C to reduce disulfide bonds. Reduced samples were then incubated with 40 mM iodoacetamide for 30 min at room temperature in the dark. By a 10-fold dilution of 12% phosphoric acid, acidified samples were loaded onto S-Trap mini columns (ProtiFi, Farmingdale, NY, USA; Cat. No: CO2-mini-80). We treated suspension-trapping (S-trap) proteolysis according to the manufacturer’s protocol, followed by the addition of 10 μg Lys-C/trypsin mixture and incubation for 16 h at 37 °C [59]. The eluted peptide mixture was lyophilized using a cold trap and stored at −80 °C until use.

### 2.3. Nano-LC-ESI-MS/MS Analysis

The LC system was an Dionex UltiMate 3000 RSLCnano system (Thermo Fisher Scientific, Waltham, MA, USA). Mobile phase A was 0.1% formic acid and 5% DMSO in water and mobile phase B was 0.1% formic acid, 5% DMSO and 80% acetonitrile in water. Samples were reconstituted with 25 µL of mobile phase A, injected with a full sample loop injection of 5 µL into a C18 Pepmap trap column (20 × 100 μm i.d., 5 μm, 100 Å; Thermo Fisher Scientific), and separated in Acclaim™ Pepmap 100 C18 column (500 × 75 μm i.d., 3 μm, 100 Å; Thermo Fisher Scientific) over 200 min (250 nL/min) at 50 °C. The column was priory equilibrated with 95% mobile phase A and 5% mobile phase B. A gradient of 5–40% B for 150 min, 40–95% for 2 min, 95% for 23 min, 95–5% B for 10 min, and 5% B for 15 min were applied. The LC system was coupled to a Q Exactive plus mass spectrometer (Thermo Fisher Scientific) with a nano-ESI source. The instrument was operated in the data-dependent mode. One scan cycle included one MS1 scan at a resolution of 70,000 at *m*/*z* 400 followed by 20 MS2 scans in higher energy collisional dissociation mode to fragment the 20 most abundant precursor ions identified in the MS1 spectrum. The target value for MS1 by Orbitrap was 3 × 10^6^ with a maximum injection time of 100 ms. The ion target value for MS2 was set to 1 × 10^6^ with a maximum injection time of 50 ms and a resolution of 17,500 at *m*/*z* 400. The dynamic exclusion was enabled with the following settings: repeat count = 1 and exclusion duration = 20 s. All MS data were deposited in the Proteomics Identification Database (PRIDE) archive under PXD028251 [60].

### 2.4. Protein Identification by Database Search

Individual raw files acquired MS analysis and were retrieved against the reviewed Human Uniprot-SwissProt protein database (released on May 2017) [61] using the SEQUEST-HT on Proteome Discoverer (Version 2.2, Thermo Fisher Scientific). Search parameters used were as follows: 10-ppm tolerance for precursor ion mass and 0.02 Da for fragmentation mass. Trypsin peptides tolerate up to two false cleavages. Carbamidomethylation of cysteines was set as fixed modification and N-terminal acetylation and methionine oxidation were set as variable modifications. The false discovery rate (FDR) was calculated using the target-decoy search strategy, and the peptides within 1% of the FDR were selected using the post-processing semi-supervised learning tool Percolator [62] based on the SEQUEST result. Label-free quantitation (LFQ) of proteins was calculated using the precursor ion peak intensity for unique and razor peptides of each protein and excluded peptides with methionine oxidation.

### 2.5. Differential Data Analysis by Normalization and Filling Missing Data

The normalization method by endogenous normalization proteins, which is mainly used in LFQ, was performed [63,64]. In this study, four proteins (C6, HPX, KNG1, and SERPINC1) were selected by NormFinder software [65] because the difference between the pCR and non-pCR group was the smallest. Since the quantitative values of the four proteins depend on the characteristics of the plasma sample, they were scaled by dividing the median values of the corresponding proteins in all samples. After that, the geometric mean of the adjusted ratio values of the four proteins for each sample is calculated, and this is defined as the normalization scaling factor (NSF) for that sample. The normalized quantitative values of the remaining proteins except for the four proteins were derived by dividing the raw protein quantitative values in each sample by the NSF. The details of this method are described in previous studies [66,67,68].

Proteins were selected based on >80% of quantified proteins in all samples, and the missing data were filled by the local least squared imputation method, calculating through correlation with 100% quantified proteins at the raw abundance [69]. After that, normalization was performed.

### 2.6. Statistical Clinical Model Generation Based on Feature Selection

Feature selection was performed to identify the optimal subset from four proteins [apolipoprotein C3 (APOC3), endoglin (ENG), mannose-binding lectin 2 (MBL2), and prolyl 4-hydrolase beta (P4HB)] to classify patients into two NCT response groups using a random forest (RF)-based backward elimination process [70]. This process consisted of the following two steps: first, 10,000 decision trees containing four variables were randomly generated, and area under the curve (AUC) values were calculated. The AUC values were used to determine the optimal number of proteins using out-of-bag error estimation, yielding a value of three. Second, 50 iterations and three-fold cross-validation were performed using the three selected variables to calculate the predictive importance of each variable included in the model. Three proteins (>0.5 probability of selection) were selected. Data preprocessing, including centering and scaling, were performed before model building. The training and validation sets were divided into thirds using whole data. In the training set, a linear kernel support vector machine (SVM) model [71] with optimized cost parameters was generated by three-fold cross-validation with three repeats. The RF model [72] was optimized with a mtry parameter by three repeats of three-fold cross-validation with 10,000 trees and nodeSize = 5. Machine learning (ML) model prediction values were obtained in the validation set (without the training set samples), and ROC analysis was performed. This process was performed by randomly changing sets 100 times.

### 2.7. Mining Public Microarray Data

Microarray gene expression data (series accession number: GSE22513 [73,74,75] and GSE22093 [76,77]) were downloaded from the Gene Expression Omnibus database [78]. The GEO2R interactive web tool was used to extract three identifiers that matched the three selected genes according to the platform record and their expression values. When there were two or more probes for one gene, the median value was estimated.

### 2.8. Statistical Methods

Survival analyses were performed including DFS, DMFS, and OS. DFS was defined as the time from the date of enrollment into the study to the first date of any type of recurrence. DMFS and overall survival (OS) were defined as the time elapsed between the date of enrollment into the study and the date of distant metastasis or the date of death from any cause, respectively. Statistical analysis was performed using IBM SPSS Statistics Version 26. The univariate Kaplan–Meier method was used to estimate survival probabilities. Multivariate Cox proportional hazards regression analyses were performed for each proteome using the following clinical parameters: patient age at diagnosis; clinical tumor stage; nodal status; hormone receptor (HR) status; and HER2 status. A *p*-value of <0.05 was considered to be statistically significant.

Proteome data evaluation was performed by statistical language R 3.6.0 and RStudio 1.1.456 with the several packages that contained ggplot2 for displaying violin and volcano plots, mixOmics for PLA-DA [79], RVAideMemoire for calculating VIP scores, stats for applying the t-test, pcaMethods for missing data imputation, survival for survival analysis, survminer for determining cutoff values by maximally selected rank statistics (minimal proportion of observations per group: 20%), and GEOquery for downloading GEO sets.

## 3. Results

### 3.1. Baseline Characteristics

The patient characteristics of each subgroup and the NCT regimen of the patients are summarized in Table 1 (detailed in Appendix A). Twenty-eight patients (47.5%) were HR+/HER2-, five patients (8.5%) were HR+/HER2+, six patients (10.2%) were HER2+, and 20 patients (33.9%) had TNBC. The mean age was 59 years (range, 32–66 years; median age, 46 years). Thirty-eight patients (74.5%) were lymph node-positive. TNBC patients had a significantly higher tumor nuclear grade (Grade 3, 57.1%). Forty-seven patients (92.2%) underwent chemotherapy with an anthracycline-based regimen. According to RECIST criteria [57], 49 patients (96.1%) demonstrated a partial or complete response (PR or CR), two patients (3.9%) showed progressive disease, and none showed stable disease. Twenty-six of the 51 patients underwent breast-conserving surgery followed by radiation therapy (100.0%, 26/26), and 25 had a mastectomy. Of those 25 patients, 20 (80.0%) patients with tumor stage ≥3 or nodal stage ≥2 selectively received radiation therapy. All HR-positive patients received hormonal therapy after surgery. Fifteen patients (29.4%, 15/51) achieved pCR. The pCR rate was significantly higher in HER2+ or triple-negative tumors (HER2+, 40%, 2/5; HR+/HER2+, 60%, 3/5; and triple-negative, 38.1%, 8/21) than in HR+/HER2- patients (10.0%, 2/20), consistent with previous studies [12,80].

### 3.2. Proteome Results from Clinical Plasma Samples by LC-MS/MS

A workflow was established for biomarker identification in BC patients with or without pCR after NCT (Figure 1A). To identify prognostic marker candidates for pCR, clinical plasma samples were collected from 51 BC patients, including 15 with and 36 without pCR after NCT. Depleted plasma samples from the 51 study participants were used to analyze constitutive proteins via single LC-MS/MS runs, which led to the identification of 594 proteins. Among them, 548 proteins were quantified in one or more samples using a label-free quantification method. Among these, four relatively stable abundant proteins (C6, HPX, KNG1, and SERPINC1) were used to normalize the raw abundance of the other candidates, which were quantified in at least 80% of the samples [81] (Appendix A). Before normalization, missing values were imputed [82]. After normalization, 254 common proteins out of 305 proteins showed a significant positive correlation with the plasma concentrations of the published Plasma Proteome Database [83] (*ρ* = 0.657; Pearson’s correlation coefficient, permutation *p* < 0.001; Appendix A). Partial least squares-discriminant analysis (PLS-DA) indicated that the pCR and non-pCR groups were separated into two components, component 1 (6%) and component 2 (17%) (Figure 1B). VIP score-ordered contributions are shown in Figure 1C. The top 26 proteins had VIP scores >1.5. Statistical analysis was performed to identify pCR prediction marker candidates. Four signature proteins annotated by molecular functional terms and processes were selected for building the clinical model. Finally, three biomarkers were selected and used in the survival analysis, including recurrence, death, and metastasis events.

### 3.3. Differentially Abundant Plasma Proteins between pCR and Non-pCR BC Patients

Statistical analysis was performed using the Student’s t-test to identify differentially abundant plasma proteins (DAPs) between the two groups. A volcano plot was drawn to represent log2 fold-changes against negative log10 *p*-values. We identified a single upregulated protein in the pCR group and three proteins in the non-pCR group (*p* < 0.05 and |fold-change| > 2; Figure 2A and Appendix A). We examined whether the abundance of the four proteins was related to the subtypes as confounding factors. Each protein was stratified by pCR status, and the quantitative differences according to subtype (HER2 and HR positive or negative) were statistically analyzed (*p* > 0.05; Appendix A). The results confirmed that the subtype did not affect the four proteins as a confounding factor.

Functional annotation of the proteins was performed using Enrichr [84]. Significant differences between the two groups were identified using WikiPathways (Figure 2B). P4HB and ENG were involved in the “VEGFA-VEGFR2 signaling pathway”. ENG was also involved in “transforming growth factor beta binding” and “hypothesized pathways in pathogenesis of cardiovascular disease”. P4HB was also associated with “type I collagen synthesis in the context of osteogenesis imperfecta”. MBL2 was related to “complement system”, “Ebola virus pathway on host”, and “regulation of toll-like receptor signaling pathway”. APOC3 was linked to “PPAR signaling pathway”, “composition of lipid particles”, and “statin inhibition of cholesterol production”. In addition, we focused on the TNBC subtype, which shows the greatest long-term clinical benefit from pCR in BC [85]. Statistical analysis was performed as described above by dividing patients into pCR and non-pCR groups only in the TNBC subtype (Appendix A). In all BC patients, one highly abundant protein, MBL2, was identified in the non-pCR group, and three highly abundant proteins, DCD, KNG1-2, and TLN1, were identified in the non-pCR group. Two highly abundant proteins, ALCAM and MAN1A1, were identified in the pCR group.

### 3.4. Multivariate Analysis for Predicting pCR Outcome

Multivariate analysis was performed using ML classifiers based on random forest (RF) [72] and SVM [71] to improve the predictive performance for distinguishing pCR from non-pCR patients. First, feature selection was performed with the four significant proteins by AUC-based RF backward elimination [70] according to a probability of selection >0.5 (Table 2) and independently performed 305 proteins as input (Appendix A). The RF and SVM models were built with three proteins (MBL2, ENG, and P4HB). To avoid overfitting, threefold cross-validation was performed three times to generate 10,000 decision trees from the RF model, and a linear SVM model was applied. To confirm the robustness of the ML models, the sample was randomly trisected 100 times, and the model was then built with 2/3 of the sample and validated with 1/3 of the sample. Evaluation of the performance of the classifiers showed that the median AUC values for SVM and RF were 0.861 (95% CI: 0.845–0.873) and 0.861 (95% CI: 0.830–0.867), respectively (Figure 3).

The three plasma biomarkers were also expressed in tissues of BC patients. The mRNA expression levels of five proteins obtained from BC tissues by fine needle aspiration prior to NCT were analyzed. The data were obtained from two publicly available GEO datasets (GSE22513 [73,74,75] and GSE22093 [76,77]). ML models were built as described above. In GSE22513, the median AUC values for SVM and RF were 0.631 (95% CI: 0.613–0.643) and 0.646 (95% CI: 0.633–0.669), respectively. In GSE22093, the median AUC values for SVM and RF were 0.709 (95% CI: 0.684–0.713) and 0.658 (95% CI: 0.645–0.666), respectively.

### 3.5. Survival Analysis

During a median follow-up of 52.0 months, 15 relapses (29.4%) and eight deaths (15.7%) were observed. To determine the correlation of single plasma proteins with long-term clinical indicators such as DFS, OS, and DMFS, we performed univariate survival analysis for the three proteins in the model (MBL2, ENG, and P4HB), and the remaining 302 proteins were quantified. The Kaplan–Meier method was used to select the cutoff values based on the maximally selected rank statistics. At first, pCR was statistically a better prognostic factor than non-PCR for DFS, OS, and DMFS (log-rank test *p* < 0.05). MBL2 and P4HB for DFS, P4HB for OS, and MBL2 for DMFS were statistically significant in dividing patients into low-risk and high-risk groups (log-rank test *p* < 0.05; Figure 4, Appendix A). Among the remaining 302 proteins quantified, the prognosis with respect to the three survival results in the two patient groups was separated by a threshold of protein quantification values: 84 proteins for DFS, 46 proteins for OS, and 96 proteins for DMFS were statistically significant for classifying patients into high-risk and low-risk groups (log-rank test *p* < 0.05; Appendix A). We also analyzed the prognosis of patients according to Miller-Payne grades in the patients with partial response, but couldn’t find a significant correlation (Appendix A). In the multivariate Cox analysis of survival with following factors: patient age at diagnosis; clinical tumor stage; nodal status; hormone receptor (HR) status; and HER2 status, no factor showed significant correlation. However, in DMFS analysis with proteins, MBL2 was identified as the only consistent risk factor (HR: 9.65, 95% CI: 2.10–44.31, *p* = 0.004; Table 3). In other survival analyses including DFS and OS, none of the proteins demonstrated a significant correlation. All three protein (MBL2, ENG, P4HB) levels were significantly increased as the pathological stages elevated (Appendix A).

## 4. Discussion

Despite considerable advances in our understanding of BC biology, the design of therapeutic approaches is dependent on and guided by molecular profiling that categorizes tumors according to HR and HER2 status [86]. Despite the improvement of treatment strategies related to HR and HER2 status, recent emerging global trends show increased BC mortality rates [87], which are attributed to treatment resistance and highly proliferative BC variants within these subtypes [88]. Thus, the identification of novel markers that can detect resistance and prognosis is an important issue.

The pCR can be a potential surrogate marker with a prognostic value for predicting survival in the HER2-positive and triple-negative subtypes [11,89,90,91]. In these subtypes, patients who achieve pCR have a better prognosis than those who fail to achieve pCR [10,11,12]. Thus, predicting tumor response before or during NCT is important for evaluating patients’ prognosis. However, in luminal subtype BC, reliable factors associated with tumor response and prognosis are relatively rare.

In luminal subtype BC, a genomic assay based on tissue biopsy samples such as the Oncotype DX^®^ provides information on clinicopathological factors and is a powerful precision medicine tool for cancer patients. However, its application is limited to the HR-positive and HER2-negative BC subtypes [14]. Moreover, the assay requires a relatively large specimen and is limited by high costs and long processing times.

Another strategy to obtain information for tumor assessment is a ‘liquid biopsy’. Compared with tissue biopsy, a quick and easy ‘liquid biopsy’ allows for a less invasive and simpler assessment of tumor response through simple blood collection [19]. Plasma cancer biomarkers and the number of circulating cancer cells provide useful information that is relevant to cancer diagnosis. CTCs are potential biomarkers of prognosis after treatment [24,26,92,93]. Although studies show that CTC detection is a potential prognostic factor in metastatic BC [21,22,94,95], its clinical value and prognostic impact in non-metastatic BC patients, especially those treated with NCT, are under debate with conflicting results [23,92,96,97].

Proteomics is the analysis method that is similar to CTC analysis and also another form of “non-invasive analysis” as liquid biopsy. Proteomic profiles derived from liquid biopsy samples can provide more direct profiling of diseases. Thus, proteomics is expected to yield promising results for the early diagnosis of cancer and for evaluating the efficacy of antitumor therapy. Current clinical proteomics methods for cancer management are focused on biomarker discovery and validation [98]. Because proteins function through specific pathways rather than individually, anti-cancer strategies can be designed by targeting the biomarker that affects specific pathways related to cancer development [99,100].

In the present study, we analyzed the plasma proteomes of locally advanced BC patients receiving NCT. We used a biomarker discovery workflow system to identify candidate protein biomarkers of tumor response and prognosis in BC patients using LC-MS/MS and three proteins were selected for validation. These proteins were used to generate models that independently predicted the risk of recurrence regardless of tumor subtype. The three protein markers were P4HB, ENG, and MBL2. P4HB is a protein disulfide-isomerase that is one of the core genes in the beta subunit of prolyl 4-hydroxylase [101]. P4HB is related to the carcinogenesis and development of multiple tumors. P4HB is highly expressed in colon cancer, and knockdown of P4HB promotes cancer cell apoptosis [102]. In liver cancer, knockdown of P4HB inhibits the migration and invasion of HepG2/ADR cells, as demonstrated by culturing HepG2 cells (a human hepatocellular carcinoma cell line) in the presence of increasing concentrations of adriamycin [103]. P4HB is also highly expressed in renal clear cell carcinoma and significantly related to poor OS [104]. These findings indicate that P4HB may serve as a potential molecular marker for the diagnosis and treatment of cancer. We demonstrated that low serum P4HB level is significantly associated with better DFS and OS, which is consistent with the findings reported by Yang et al. [105]. The study demonstrated that downregulation of P4HB represses the promoting effects of overexpressed COL10A1 on the proliferation, migration, and invasion of BC cells and, conversely, upregulation of P4HB promotes BC cell proliferation and clone-forming ability, as well as increasing BC cell migration and invasion [105].

ENG (also known as CD105) is a receptor for transforming growth factor β that is expressed at high levels on the cell surface of tumor blood vessels and tumor stromal components [106]. ENG shows affinity for “newly forming” angiogenic endothelium, whereas CD34 and CD31 react not only with angiogenic vessels, but also with the endothelium of normal vessels; ENG is thus superior to CD34 and CD31 for the evaluation of tumor angiogenesis [107]. Elevated expression of ENG is often observed in the actively proliferating endothelium [108,109]. There is a significant correlation between markers of cell proliferation such as Ki-67 and cyclin-A [108]. Thus, ENG is a potential marker of tumor-associated angiogenesis and prognosis [109]. Li et al. showed that plasma ENG levels are elevated in BC patients at risk of metastasis, and ENG overexpression is significantly correlated with metastatic disease, suggesting the value of ENG for predicting metastasis [110]. Kumar et al. demonstrated that the reactivity of ENG in blood vessels of BC tissues correlates with a poor prognosis [111]. Although the results did not reach statistical significance, we also observed that patients with high ENG levels had poor survival rates.

MBL2 is an activator of the lectin pathway and a crucial component of the innate immune system, and inflammatory reactions are critical for tumor progression and can promote human carcinogenesis [112,113]. MBL can inhibit tumor progression via the complement system and through MBL-dependent cell-mediated cytotoxicity [114,115,116]. However, recent studies show controversial results. Holm et al. showed that high plasma levels of MBL2 are a marker of poor survival in colorectal cancer patients [117]. Yitting et al. reported that the MBL complement activation pathway is activated in patients with colorectal cancer compared with healthy controls; however, MBL pathway deficiency rates are similar between patients and healthy controls [118]. Additionally, local expression of MBL2 genes is higher in women with ovarian cancer than in controls [119]. In this study, plasma MBL2 levels were higher in the non-pCR group, and high plasma MBL2 levels were associated with poor survival (DFS and DMFS). Taken together, these results suggest that MBL2 can have a protective effect against tumors, as well as a tumorigenic effect.

The present study had several limitations. First, the sample size was small, which limits the statistical power. To overcome any possible overfitting issue associated with the small sample size, cross-validation was used for model development. In addition, the validation study included additional plasma samples collected under IRB. Second, because the study sample was heterogeneous regarding the distribution of tumor subtypes, representing the general BC population is difficult. However, considering the prevalence of each subtype of BC, the present results are valuable for discovering prognosis-related protein signatures. Third, although tumor-derived proteomes are present at high concentrations in the blood of cancer patients, abundant proteins can be derived from cellular sources other than the tumor, which is the major limitation of “liquid biopsy”. In terms of homeostasis, the plasma proteome can reflect differences in the immune or inflammatory status of patients, which may affect the response to chemotherapy. The plasma proteome is thus a critical indicator of the chemotherapy response.

Despite the limitations listed above, the findings of this study show that certain proteomes are associated with chemotherapy response and prognosis in patients receiving NCT.

## 5. Conclusions

This study demonstrated that proteins from non-invasive liquid biopsy sampling correlate with pCR and survival in BC patients receiving NCT. Among them, potential druggable targets were identified. Plasma protein analyses identified differentially expressed proteins between groups with distant metastasis, independently from the achievement of pCR. Quantitative protein analyses by liquid biopsy may provide a means to predict response and recurrence with minimal amounts of sample, at a lower cost, and with faster times. Further investigation of these proteomes may reveal their role in predicting prognosis, which could serve as a novel therapeutic strategy.

## Figures and Tables

**Figure 1 cancers-13-06267-f001:**
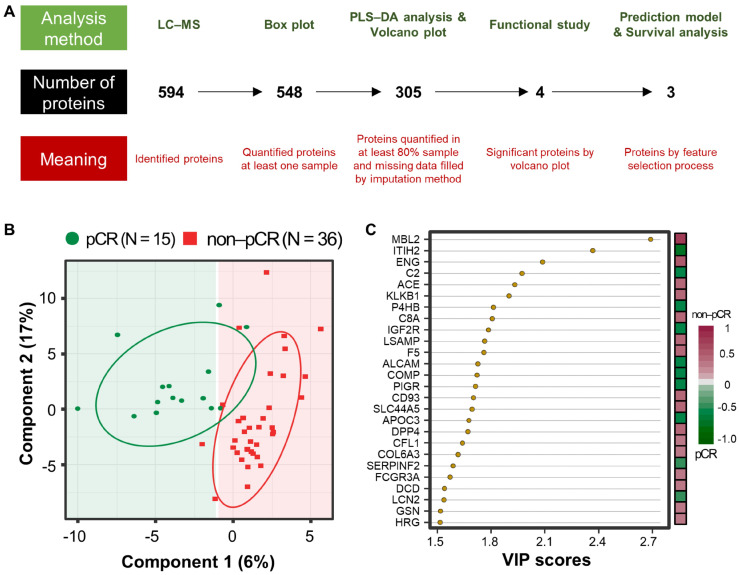
Analysis workflow and partial least-squares discriminant analysis (PLS-DA) of plasma proteomes in 51 breast cancer (BC) patients. (**A**) The analysis method is shown at the top, the number of proteins is shown in the middle, and the meaning of the step is shown at the bottom. PLS-DA score plot (**B**) and top26 variable importance in projection (VIP) score (>1.5) plot derived from PLS-DA analysis (**C**) in 15 patients with pathological complete response (pCR; green) after neoadjuvant therapy and 36 patients with non-pCR (red).

**Figure 2 cancers-13-06267-f002:**
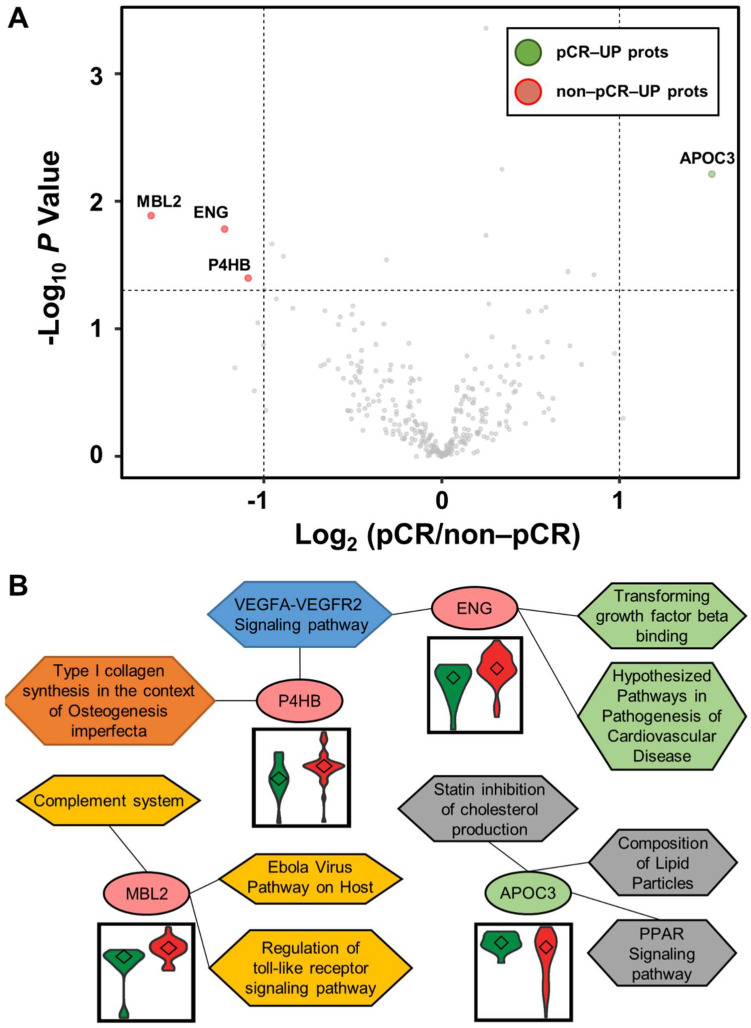
Volcano plot for DAPs altered by pathological complete response (pCR) and functional interpretation. (**A**) Log_2_ fold changes and the corresponding *p*-values of all proteins between pCR group (*n* = 15) and non-pCR group (*n* = 36) are presented as volcano plot. Proteins upregulated with more than a twofold change with a *p*-value < 0.05 are depicted in red circles and those downregulated with identical fold change and *p*-value are in green circles. Gray circles show plasma proteins that did not show statistically significant differences. (**B**) Association between WikiPathways and proteins, and violin plots of the corresponding proteins between the two groups (pCR: green; non-pCR: red).

**Figure 3 cancers-13-06267-f003:**
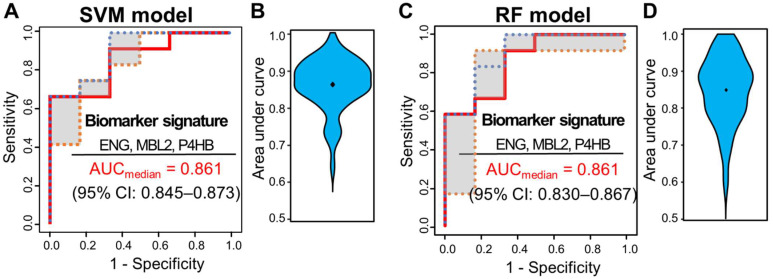
ROC curves of SVM and RF classifiers for three selected proteins (ENG, MBL2, and P4HB). Capability of the two classifiers in a set of 51 samples, 15 from patients with pCR and 36 from patients with non-pCR. (**A**) ROC curves of SVM classifiers generated through 100 repeats of threefold cross-validation steps. (**B**) ROC curves of SVM classifiers generated through 100 repeats of threefold cross-validation steps. ROC curves were obtained by plotting the 25th, 50th, and 75th quantiles of the sensitivities for each value of 1-specificity. (**B**) Violin plots of 100 area under the curve (AUC) values in the SVM model. (**C**) ROC curves of RF classifiers generated through 100 repeats of threefold cross-validation steps. ROC curves were obtained by plotting the 25th, 50th, and 75th quantiles of the sensitivities for each value of 1-specificity. (**D**) Violin plots of 100 AUC values in the RF model.

**Figure 4 cancers-13-06267-f004:**
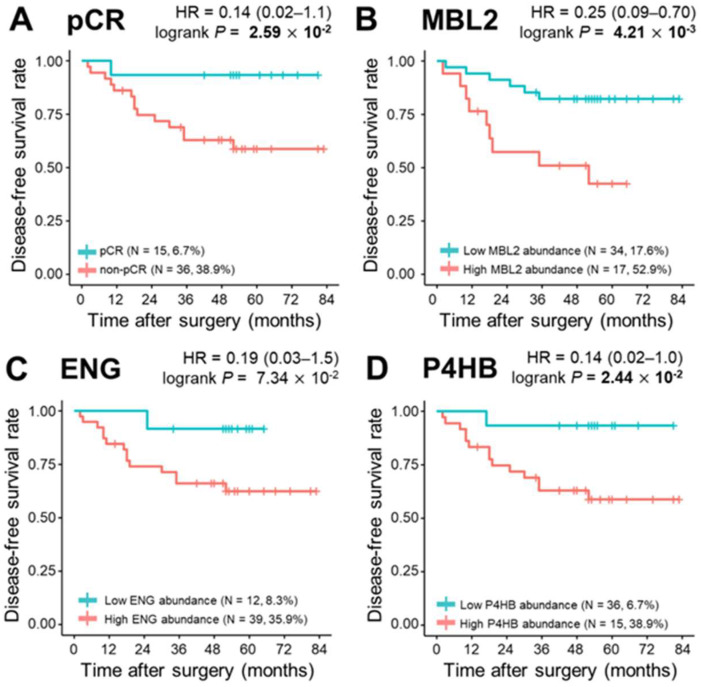
Kaplan–Meier plots of pathological complete response (pCR) and three proteins, MBL2, ENG, and P4HB. (**A**) Categorization of patients into pCR and non—pCR risk groups (pCR, *n* = 15, 6.7%; non-pCR, *n* = 36, 38.9%; *p* = 2.59 × 10^−2^). Classification of patients into risk groups according to (**B**) MBL2 abundance: low abundance group (*n* = 34, 17.6%) and high abundance group (*n* = 17, 52.9%), *p* = 4.21 × 10^−3^; (**C**) ENG abundance: low abundance group (*n* = 12, 8.3%) and high abundance group (*n* = 39, 35.9%), *p* = 7.34 × 10^−2^; and (**D**) P4HB abundance: low abundance group (*n* = 36, 6.7%) and high abundance group (*n* = 15, 38.9%), *p* = 2.44 × 10^−2^. Statistical significance was determined using the log-rank test. *p*-values < 0.05 are displayed in bold.

**Table 1 cancers-13-06267-t001:** Patient characteristics and the NCT regimen.

Variables		HR+/HER2- (*n* = 20)	HR+/HER2+ (*n* = 5)		HER2+ (*n* = 5)		Triple-Negative (*n* = 21)	*p*
Age at diagnosis (range)		32–58	41–66		45–59		35–53	0.463
≤40		18 (90.0%)	3 (60.0%)		4 (80.0%)		17 (81.0%)	
>40		2 (10.0%)	2 (40.0%)		1 (20.0%)		4 (19.0%)	
Clinical T stage								0.206
T1		0 (0%)	0 (0%)		0 (0%)		0 (0%)	
T2		11 (55.0%)	2 (40.0%)		5 (100%)		15 (71.4%)	
T3		8 (40.0%)	2 (40.0%)		0 (0%)		6 (28.6%)	
T4		1 (5.0%)	1 (20.0%)		0 (0%)		0 (0%)	
Lymph node status								0.473
Negative		6 (30.0%)	0 (0%)		2 (40.0%)		5 (23.8%)	
Positive		14 (70.0%)	5 (100%)		3 (60.0%)		16 (76.2%)	
Nuclear grade								0.001
G1 and G2		19 (95.0%)	5 (100%)		4 (80.0%)		9 (42.9%)	
G3		1 (5.0%)	0 (0%)		1 (20.0%)		12 (57.1%)	
Tumor response (RECIST)								0.295
CR		3 (15.0%)	1 (20.0%)		0 (0%)		7 (33.3%)	
PR		17 (85.0%)	4 (80.0%)		5 (100%)		12 (57.2%)	
SD		0 (0%)	0 (0%)		0 (0%)		0 (0%)	
PD		0 (0%)	0 (0%)		0 (0%)		2 (9.5%)	
Type of Surgery (adjuvant RT)								0.419
BCS (26/26)		8 (40.0%)	4 (80.0%)		3 (60.0%))		11 (52.4%)	
Mastectomy (20/25)		12 (60.0%)	1 (20.0%)		2 (40.0%)		10 (47.6%)	
Pathological response								0.047
pCR		2 (10.0%)	3 (60.0%)		2 (40.0%)		8 (38.1%)	
non-pCR		18 (90.0%)	2 (40.0%)		3 (60.0%)		13 (61.9%)	
	NCT regimen							
Anthracycline based (AC#4, AC#4 > D#4, FEC#4 > D#4)						47 (92.2%)		
NCT02032277 * (Veliparib/Placebo + Carboplatin/Placebo + Paclitaxel)						4 (7.8%)		

AC: adriamycin and cyclophosphamide, BCS: breast conserving surgery, CR: complete response, D: docetaxel, FEC: fluorouracil, epirubicin, and cyclophosphamide, N/A: not applicable, NCT: neoadjuvant chemotherapy, pCR: pathological complete response, PD: progressive disease, PR: partial response, RECIST: Response Evaluation Criteria in Solid Tumors, SD: stable disease. * The results of the trial have not yet been reported.

**Table 2 cancers-13-06267-t002:** Selected feature proteins by AUC-based RF backward elimination.

Uniprot Accession No.	Gene Name	Importance	Prob. Select *	Selection	Univariate AUC
P11226	MBL2	6.105	0.96	Y	0.807
P17813	ENG	5.556	0.85	Y	0.739
P07237	P4HB	3.522	0.58	Y	0.722
P02656	APOC3	NA	NA	N	0.654

* Probability of selection for each variable.

**Table 3 cancers-13-06267-t003:** Multivariate Cox analysis of DMFS including the MBL2 proteome.

	DMFS	
	Multivariate HR (95% CI)	*p*
Patient age (>40 vs. ≤40)	1.30 (0.33–5.06)	0.709
Tumor size (≤5 cm vs. >5 cm)	2.55 (0.56–11.65)	0.226
Node negative vs. positive	* 2.1 × 10^5^	0.963
HR positive vs. negative	2.95 (0.63–13.88)	0.172
HER2 negative vs. positive	0.61 (0.07–5.44)	0.660
MBL2 abundance (low vs. high)	9.65 (2.10–44.31)	0.004

Patients were divided into two risk groups according to MBL2 abundance: low abundance group (*n* = 34, 17.6%) and high abundance group (*n* = 17, 52.9%); HR, hormone receptor. * No events in the node negative group.

## Data Availability

The raw data supporting the conclusions of this article will be made available by the authors, without undue reservation.

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
