# Peer review of "Plasma Proteome Signature to Predict the Outcome of Breast Cancer Patients Receiving Neoadjuvant Chemotherapy"

_cancers, 2021, doi:10.3390/cancers13246267_

Round 1
Reviewer 1 Report
The manuscript entitled "Plasma Proteome Signature to Predict the Outcome of Breast Cancer Patients Receiving Neoadjuvant Chemotherapy" shows the exploration and analysis of plasma biomarkers for predicting the outcome of breast cancer patients undergoing neoadjuvant chemotherapy. The result has a potential impact. However, there are several issues that need to be resolved.
1.The sample size of 51 patients is too small, and a cohort with a larger number of patients is needed to validate the discovery in the article.
2.The inclusion and exclusion criteria of patients should be described in more detail.
3.In the patients with PR, the prognosis of patients with high MP grades (Grade 4) is different from low MP grades (Grade 1-3). A larger cohort including patients with both enough low and high MP grades should be used to validate the discovery in the manuscript.
4.The value of pCR rate in breast cancer patients is controversial, therefore, HR+/HER2- patients should be analyzed separately.
5.The association between the biomarkers with pathological stages and grades should be included.
Reviewer 2 Report
Review on Gwark et al. “Plasma proteome signature to predict the outcome of breast 2 cancer patients receiving neoadjuvant chemotherapy”
The authors analysed 51 plasma samples from patients suffering from breast cancer of different subtype and stage in order to find potential biomarker with different abundance in patient groups with positive response or non-responders upon a neoadjuvant therapy. Furthermore, selected proteins were investigated for their potential to predict disease progression and survival. The topic is relevant, since plasma derived biomarker could support an individualized treatment and/or surveillance strategy.
Major concerns:
- In general, study, methods and results are not clearly structured and the contents of the different parts not well coordinated. Hence, the abstract and the results part do not fit to each other (l. 45 “biomarkers were identified…”: which comparison?; MBL is the only candidate passing multivariate analysis-not mentioned in the abstract; etc) .
- Protein numbers named in the text are confusing: in the figure and later in the text it is mentioned that 305 proteins were quantified and used for the statistical test. However, in line 284 it is stated that “548 proteins were quantified in one or more samples, and 182 were obtained using a label-free quantification method”. Where does the number of 182 come from? Are these proteins quantified in 100% of all samples? Is there an explanation why there is such a huge difference between the quantified proteins in at least one sample and proteins quantified in 80% of samples? Is there a biological meaning behind? What is the impact of data imputation on normalization? Normalization after imputation of missing values is rather unusual.
- The authors focus very early on a four protein panel derived from a simple student’s T-Test and start Random forest and SVM classification with these 4 proteins only which all leads to overfitting of the analysis. This unusual workflow reveals several questions:
- For the binary comparison: What about the result of a logistic regression analysis for the therapy response considering covariates as age and hormone status? Authors review the impact of potential confounders only after selecting the protein candidates.
- Random forest is a tool to select predictive signatures without any preselection of variables. What is the result when all proteins are used to feed the model?
- 225-230 Here it remains unclear if SVM was part of the RF model or used as a separate toll to analyses data. From figures the latter case can be assumed, but the description is mixed up with the RF parameters.
- Authors state that they did pathway enrichment analyses, but this does not make sense when a panel of 4 proteins is analysed. Rather the authors used the analysis tool EnrichR to learn to which functional categories the proteins are assigned. This should be clearly stated and not overinterpreted as it is currently in section 3.3
- The low robustness of results becomes visible in the survival analysis part, where only MBL2 passes the significance value of p<0.05 when other covariates are considered. Due to the low number of samples analysed-which were also used for identification of biomarker candidates-and the low significance value (no candidate passes multiple test correction) the validation of results on an independent test set is necessary. The authors mention a validation cohort of additional plasma samples, but it is not clear which samples and results this statement relates to.
Minor concerns:
- Circulating proteins in the blood can have differential levels, but the expression of the genes and the protein synthesis does not occur in blood plasma, Therefore, the term “up- or down-regulated protein expression” should be strictly avoided in this context.
- The term “liquid biopsy” is misleading and should not be used for blood derived specimen, since there also real “liquid biopsies” taken from organs like the pancreas, gut or stomach.
Reviewer 3 Report
This research was aiming to screen plasma protein biomarkers in breast cancer patients treated with NCT. With LC system and analysis, MBL2, ENG and P4HB were selected in non-pCR patients and further validated using KM plots. These proteins/genes may serve as the biomarkers benefiting clinical prognosis in BCs.
Concerns
1 the number in present study only 51, this is a limitation for the conclusion.
2 the authors claim that the results roles in therapeutic potential for preventing metastasis in the Summary, however the patients are all non-metastatic breast cancer.
3 online dataset like METABRIC should be used to test the protein markers actions
Round 2
Reviewer 1 Report
For Points 1, 2, 4, and 5, the author has given the appropriate response. For point 3, the author's analysis results still need to be verified by a large number of clinical samples to avoid bias. We hope that the authors can improve it in their upcoming analyses.
Author Response
We are deeply appreciative of the insightful comments and constructive suggestions which improved our study. As we are aware of the limitation regarding point 3, we will improve it in our upcoming reports.
Reviewer 2 Report
Review on Gwark et al. “Plasma proteome signature to predict the outcome of breast 2 cancer patients receiving neoadjuvant chemotherapy”
Thanks to the authors for the revision of the manuscript and the detailed explanation of the statistical methods applied and the rationale behind.
Running RF with all proteins detected revealed new protein candidates that might be of interest for further analyses. Authors should add the results as supplementary material and make a short notice on the findings in the manuscript.
The sentence” Significant differences between the two groups were identified using WikiPathways (adjusted p < 0.05; Figure 2B).” should be deleted because no meaningful enrichment analysis can be performed with only 4 molecules.
The meaning of the added “…for clinical application” in l. 294 is not clear, because “samples” refer to” analysed samples”.
- 442 Proteomics is not a form of non-invasive sampling, but rather an analysis method. Please revise the sentence
Please revise remaining spelling errors that should be corrected:
- 134 “Asian…”
- 453 delete “.” at “LC-MS/MS”
Table 1 and footnote and l. 545 “Pathologic” –> “Pathological”
Author Response
We thank the reviewers for the insightful comments and constructive suggestions. We have considered all comments and suggestions and revised the manuscript accordingly. Please see below for a point-by-point response to each of the points made by the reviewers. Page numbers listed below are based on the revised manuscript version.
Point 1: Running RF with all proteins detected revealed new protein candidates that might be of interest for further analyses. Authors should add the results as supplementary material and make a short notice on the findings in the manuscript.
Response 1: We agree with this reviewer’s comment. We made Supplementary Table S4 and wrote the contents in the revised manuscript at line 352-353.
Point 2: The sentence” Significant differences between the two groups were identified using WikiPathways (adjusted p < 0.05; Figure 2B).” should be deleted because no meaningful enrichment analysis can be performed with only 4 molecules.
Response 2: We agree with this reviewer’s comment. We modified the context from “(adjusted p < 0.05; Figure 2B).” to “(Figure 2B).” in the revised manuscript at line 325-326.
Point 3: The meaning of the added “…for clinical application” in l. 294 is not clear, because “samples” refer to” analysed samples”.
Response 3: We agree with this reviewer’s comment. We removed the words “for clinical application” in the revised manuscript at line 294.
Point 4: 442 Proteomics is not a form of non-invasive sampling, but rather an analysis method. Please revise the sentence
Response 4: We agree with this reviewer’s comment. We modified the context from “non-invasive sampling” to “non-invasive analysis” in the revised manuscript at line 444.
Point 5: Please revise remaining spelling errors that should be corrected:
- 134 “Asian…”
- 453 delete “.” at “LC-MS/MS”
Table 1 and footnote and l. 545 “Pathologic” –> “Pathological”
Response 5: We thank the reviewer for these comments. In number 1, ASAN is a proper noun and is the name of the founder of the institute (ASAN medical center). In number 2, we deleted the period in the revised manuscript at list 453. Revised “Pathologic” to “Pathological” in Table 1 and footnote and l. 545.